# ONLINE BLACK-BOX ADAPTATION TO LABEL-SHIFT IN THE PRESENCE OF CONDITIONAL-SHIFT

## ABSTRACT

We consider an out-of-distribution setting where trained predictive models are deployed online in new locations (inducing *conditional-shift*), such that these locations are also associated with differently skewed target distributions (*label-shift*). While approaches for online adaptation to label-shift have recently been discussed by Wu et al. (2021), the potential presence of concurrent conditional-shift has not been considered in the literature, although one might anticipate such distributional shifts in realistic deployments. In this paper, we empirically explore the effectiveness of online adaptation methods in such situations on three synthetic and two realistic datasets, comprising both classification and regression problems. We show that it is possible to improve performance in these settings by learning additional hyper-parameters to account for the presence of conditional-shift by using appropriate validation sets.

## 1 INTRODUCTION

We consider a setting where we have black-box access to a predictive model which we are interested in deploying online in different places with skewed label distributions. For example, such situations can arise when a cloud-based, proprietary service trained on large, private datasets (like Google's Vision APIs) serves several clients real-time in different locations. Every new deployment can be associated with label-shift. Recently, Wu et al. (2021) discuss the problem of online adaptation to label-shift, proposing two variants based on classical adaptation strategies – *Online Gradient Descent* (OGD) and *Follow The Leader* (FTH). Adapting the output of a model to a new label-distribution without an accompanying change in the label-conditioned input distribution only requires an adjustment to the predictive distribution (in principle). Therefore, both methods lend themselves to online black-box adaptation to label-shift, which makes on-device, post-hoc adjustments to the predictive distribution feasible under resource constraints.

In this paper, we empirically explore such methods when the underlying assumption of an invariant conditional distribution is broken. Such situations are likely to arise in reality. For example, in healthcare settings there are often differing rates of disease-incidence (label-shift) across different regions (Vos et al., 2020) accompanied by conditional-shift in input features at different deployment locations, for example in diagnostic radiology Cohen et al. (2021). In notation, for input variable $x$ and target variable $y$, we have that $P^{\text{new}}(x \mid y) \neq P(x \mid y)$ and $P^{\text{new}}(y) \neq P(y)$, for a training distribution $P$ and a test distribution $P^{\text{new}}$ in a new deployment location.

**Contributions**   Our contributions are as follows.

- We conduct an empirical study of the FTH and OGD methods introduced by Wu et al. (2021) in black-box label-shift settings with concurrent conditional-shift, a situation likely to arise in realistic deployments.

- We explore the question of how to potentially improve performance in such practical settings by computing confusion matrices on OOD validation sets, and show that adding extra hyper-parameters can contribute to further improvements.

- We reinterpret a simplified variant of FTH under a more general Bayesian perspective, enabling us to develop an analogous baseline for online adaptation in regression problems.

## 2 BACKGROUND

We begin with a brief review of online adaptation methods for label-shift for classification problems, based on the recent discussion in Wu et al. (2021). While their motivation is temporal drift in label-distributions, we consider the case where a single model is serving several clients online in different locations, each with their own skewed label-distribution that does not change even further with time. If the training set label-distribution is $P(y)$ and the label-distribution in the new location is $P^{\text{new}}(y)$, and if we assume $P^{\text{new}}(x \mid y) = P(x \mid y)$, then the following holds

$$P^{\text{new}}(y \mid x) = \frac{P(x \mid y)P^{\text{new}}(y)}{P^{\text{new}}(x)} = \frac{P(y \mid x)P(x)}{P(y)}\frac{P^{\text{new}}(y)}{P^{\text{new}}(x)} \propto \frac{P^{\text{new}}(y)}{P(y)}P(y \mid x), \tag{1}$$

i.e., the location-adjusted output distribution is simply a reweighting of the output distribution from the base underlying predictive model. Wu et al. (2021) follow along past work on label-shift adaptation by restricting the hypothesis space for $f$ to be that of re-weighted classifiers, since Eq. 1 implies that one only needs to re-weight the predictive distribution to account for label-shift. The parameter vector for this classifier is simply the vector of probabilities in $P^{\text{new}}(y)$, henceforth referred to as $\boldsymbol{p}$, and we will similarly use $\boldsymbol{q}$ to represent the training-set probability distribution, $P(y)$. Given an underlying predictive model $f$, the adjusted classifier rule is therefore given by

$$g(x; f, \boldsymbol{q}, \boldsymbol{p}) = \arg\max_{y \in [K]} \frac{\boldsymbol{p}[y]\,P_f(y \mid x)}{\boldsymbol{q}[y]}, \tag{2}$$

where $P_f(y \mid x)$ is the predictive distribution produced by an underlying base model $f$; for example, a softmax distribution produced by a neural network, and there are $K$ classes in our dataset.

### 2.1 ONLINE ADAPTATION ALGORITHMS

Wu et al. (2021) present two online updating methods to estimate $\boldsymbol{p}$ – Online Gradient Descent (OGD) and Follow The History (FTH).

If we assume knowledge of a confusion matrix for a classifier $f$ in a new location, $C^{\text{new}}(f) \in \mathcal{R}^{K \times K}$, such that $C_f^{\text{new}}[i, j] = P_{x \sim P^{\text{new}}(x|y=i)}(f(x) = j)$, then Wu et al. (2021) show that the expected error rate in this new location can be derived as a function of the label-distribution $P^{\text{new}}(y)$. If we represent $P^{\text{new}}(y)$ as a $K$-dimensional probability vector $\boldsymbol{q}^{\text{new}}$, the expected error rate is given as

$$\ell^{\text{new}}(f) = \sum_{i=1}^{K} \left(1 - P_{x \sim P^{\text{new}}(x|y=i)}(f(x) = i)\right) \cdot \boldsymbol{q}^{\text{new}}[i] = \langle \mathbf{1} - \text{diag}(C_f^{\text{new}}), \boldsymbol{q}^{\text{new}} \rangle, \tag{3}$$

where $\mathbf{1}$ is the all-ones vector. Since we have assumed no conditional-shift so far, $C_f^{\text{new}} = C_f$, i.e. the confusion matrix remains invariant under label-shift. This implies one can optimize the expected error rate in the new deployment location using a confusion matrix estimated from a large in-distribution validation set, $C_f$, in place of $C_f^{\text{new}}$ in Eq. 3.

**Online Gradient Descent (OGD)** Assuming that $\text{diag}(C_f)$ is differentiable $wrt f$, we can update $f$ to minimize the expected error rate. We would typically not be aware of the true label-distribution in the new deployment location. However, when the confusion matrix $C_f$ is invertible, we can compute an unbiased estimate of this distribution, given as $\hat{\boldsymbol{q}}^{\text{new}} = \left(C_f^\top\right)^{-1}\boldsymbol{e}$, where $\boldsymbol{e}$ is a one-hot vector for the predicted category. Using this, Wu et al. (2021) present an unbiased gradient of $\ell^{\text{new}}(f)$,

$$\nabla_f \hat{\ell}^{\text{new}}(f) = \mathbb{E}_{P^{\text{new}}}\left[\frac{\partial}{\partial f}[\mathbf{1} - \text{diag}(C_f)]^\top \cdot \hat{\boldsymbol{q}}^{\text{new}}\right]. \tag{4}$$

When the hypothesis space is restricted to the space of re-weighted classifiers $g$ (Eq. 2) this gradient is only over $\boldsymbol{p}$. Wu et al. (2021) show how we might use effective numerical methods to estimate this gradient. In the online setting, $\boldsymbol{p}$ is updated after seeing new examples, hence the $t + 1$-th gradient update is performed by computing the gradient at the current point $\boldsymbol{p}_t$, followed by a projection to the probability simplex,

$$\nabla_{\boldsymbol{p}} \hat{\ell}^{\text{new}}(\boldsymbol{p})\Big|_{\boldsymbol{p}=\boldsymbol{p}_t} = \mathbb{E}_{P^{\text{new}}}\left[\frac{\partial}{\partial \boldsymbol{p}}[\mathbf{1} - \text{diag}(C_g)]^\top \cdot \hat{\boldsymbol{q}}^{\text{new}}\right]\Big|_{\boldsymbol{p}=\boldsymbol{p}_t} \tag{5}$$

$$\boldsymbol{p}_{t+1} = \text{Proj}_{\Delta^{K-1}} \left( \boldsymbol{p}_t - \eta \cdot \nabla_{\boldsymbol{p}} \hat{\ell}^{\text{new}}(\boldsymbol{p}) \Big|_{\boldsymbol{p}=\boldsymbol{p}_t} \right), \tag{6}$$

where $\eta$ is the learning rate and Proj is the projection operator.

**Follow The History (FTH)**    The update rule for $\boldsymbol{p}_t$ in FTH is simpler and more efficient (in terms of memory and time complexity), given by

$$\boldsymbol{p}_{t+1} = \frac{1}{t} \sum_{\tau=1}^{t} \hat{\boldsymbol{q}}_{\tau}^{\text{new}}, \tag{7}$$

where $\hat{\boldsymbol{q}}_{\tau}^{\text{new}}$ is the estimate for the label distribution at the $\tau$-th iteration. Empirical evidence in Wu et al. (2021) suggests that FTH performs very competitively with OGD, and might be preferred in highly resource-constrained settings.

## 3    UNMET ASSUMPTIONS IN PRACTICE

We now consider applying the above strategies in cases where some of the assumptions in the above section are broken. While it is difficult to make conclusive theoretical statements in situations when these assumptions break, we propose some heuristics which we evaluate empirically.

### 3.1    THE ASSUMPTION OF INVARIANT $P(x \mid y)$ CAN BREAK

In realistic deployments in new locations, it is likely that along with a differently skewed label-distribution, the conditional distribution will change as well, i.e. $P^{\text{new}}(x \mid y) \neq P(x \mid y)$. In our study, we will assume that this distributional shift only takes place within the same domain, and along (potentially spuriously-correlated) non-semantic features, leaving the semantic features intact, a setting likely to be manifested in different deployment locations.

**HEURISTIC 1**    One possibility to adapt the above methods to settings with concurrent conditional-shift is to estimate the confusion matrix on an OOD validation set. Intuitively, an IID-estimated confusion matrix is likely to be over-confident, and a surrogate-OOD validation set can better reflect performance at test-time OOD settings.

**HEURISTIC 2**    We propose to add extra scaling hyper-parameters in the decision rule in Eq. 2. Specifically, we add the scaling hyper-parameters $\lambda_u$ and $\lambda_y$ before making a test prediction,

$$\tilde{g}(x; f, \boldsymbol{q}, \boldsymbol{p}) = \underset{y \in [K]}{\arg\max} \ \log P_f(y \mid x) + \lambda_u \log \boldsymbol{p}[y] - \lambda_y \log \boldsymbol{q}[y], \tag{8}$$

where we have rewritten the rule in log-space. In this formulation, $\log P_f(y \mid x) = \text{logit}[y] - Z(x)$, so we can drop the normalizing term. This results in a predictive rule that is a form of *logit-adjustment* (Menon et al., 2021). Intuitively, these hyper-parameters play the role of determining how much of the training prior to "subtract", and how much weight to assign to the pseudo-label based re-adjustment. When these magnitudes are learned on validation sets representing a combination of label-shift and conditional-shift, one can hope to further improve at novel test-time deployments.

### 3.2    CONFUSION MATRICES CAN BE NON-INVERTIBLE

Existing work on label-shift based on confusion matrices rely on a significantly large held-out validation set to estimate a robust confusion matrix. When the underlying dataset is highly class-imbalanced with several categories and limited-size validation sets, one can easily end up with a non-invertible confusion matrix. Lipton et al. (2018) suggests two main possibilities – use of a soft confusion matrix, or a pseudo-inverse. In our experiments on a large-scale realistic dataset, we find both choices to lead to degraded performance. We find that simply using an identity matrix approximation can recover some of the performance drops (see Appendix E). When using FTH with an identity $C_f$, this corresponds to simply using the pseudo-labels up to time $t$ to estimate the label-distribution. However, naively using the identity matrix in Eq. 7 might lead to a practical problem: after seeing the first data-point, $\boldsymbol{p}$ would be a one-hot vector, and thus enforce the same

prediction at the next iteration when using Eq. 2. A fix would be to use a "pseudo-count" to smooth initial conditions, which is reminiscent of Bayesian posterior updates. In the next section, we use this realization as a starting point to suggest a simpler as well as more general framework. This framework then enables us to develop an equivalent online label-shift adaptation method for regression problems.

## 4   A BAYESIAN PERSPECTIVE

If we use the vector $\boldsymbol{\alpha}$ to keep online counts of predictions, with an initialized $\boldsymbol{\alpha}_0$, such that

$$\boldsymbol{\alpha}_t[k] = \sum_{\tau=1}^{t} \mathbf{1}[\hat{y}_\tau = k] + \boldsymbol{\alpha}_0 = \mathbf{1}[\hat{y}_t = k] + \boldsymbol{\alpha}_{t-1}[k], \tag{9}$$

then using an identity confusion matrix in Eq. 7 corresponds to the following update rule,

$$\boldsymbol{p}_{t+1}[k] = \frac{\boldsymbol{\alpha}_t[k]}{\sum_{k'=1}^{K} \boldsymbol{\alpha}_t[k']}. \tag{10}$$

We recognize that this update-rule corresponds exactly to the posterior predictive distribution computed using a Categorical likelihood with a Dirichlet prior, and using a recursive rule for updating the posterior. More precisely, if we use

$$\phi \sim \text{Dir}(\boldsymbol{\alpha}), \tag{11}$$
$$y \mid \phi \sim \text{Cat}(\phi), \tag{12}$$

where $\phi \in \Delta^{K-1}$ are the parameters of the Categorical distribution, in the following update equations

$$P_t(\phi) \propto P(y_t \mid \phi) \, P_{t-1}(\phi), \tag{13}$$

$$P_{t+1}(y) = \int_\phi P(y \mid \phi) \, P_t(\phi) \, d\phi, \tag{14}$$

then we arrive at Eq. 10 using Eq. 14, and Eq. 9 using Eq. 13. See Appendix A for a derivation of Eq. 13. In practice, $y_t$ is not available to us, and we use the pseudo-label $\hat{y}_t$ instead, as in FTH.

### 4.1   EXTENSION TO REGRESSION PROBLEMS

While adaptation for regression problems has been discussed more generally (Cortes & Mohri, 2011; 2014; Zhang et al., 2013), an analogous discussion for online black-box label-shift adaptation is missing for regression. We adapt the general online update rules in Eq. 13, 14 for regression problems undergoing similar concurrent test-time distributional shifts. A natural choice is to use Gaussians to model the distributions over the continuous target variable,

$$P_f(y \mid x) \propto \exp\left(-\frac{\lambda_x}{2}\Big(y - f(x)\Big)^2\right), \tag{15}$$

$$P(y) \propto \exp\left(-\frac{\lambda_y}{2}\Big(y - m\Big)^2\right), \tag{16}$$

where $\lambda_x, \lambda_y$ are the precision parameters and $m$ is the training set mean. The parameters $\phi$ in Eq. 13 are now the mean and precision parameters for $y$ in the new deployment location. We use the Normal-Gamma distribution to model the posterior over these parameters, since this is the conjugate distribution for Gaussians with unknown mean and precision (DeGroot, 2004),

$$P(\mu^{\text{new}}, \lambda^{\text{new}}) = \mathcal{N}\left(\mu^{\text{new}} \mid \mu, \frac{1}{\kappa\lambda^{new}}\right) \text{Ga}(\lambda^{new} \mid a, b). \tag{17}$$

Combined with the Gaussian likelihood in Eq. 14, this yields $P^{\text{new}}(y)$ in the form of a Student's $t$-distribution,

$$P^{\text{new}}(y) \propto \left(1 + \frac{L}{2a}(y - \mu)^2\right)^{-\frac{2a+1}{2}}, \tag{18}$$

Train ($r = 0.99$)

Validation (opposite colors with $r = 0.75$)

Test ($r = -1.0$)

(a) Synthetic variant of the MNIST dataset constructed by using colors to correspond to sources with skewed label-distributions. The colors are flipped for validation and test with different correlation strengths, corresponding to (almost completely) reversing the label-skew at the sources at test-time.

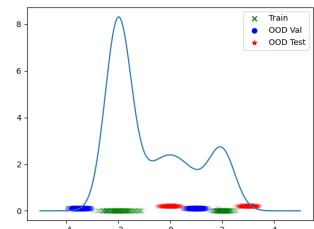

(b) Synthetic MIX-OF-GAUSSIANS data. Differently colored regions along the $x$-axis correspond to training, validation and test samples, with different regions of the same color corresponding to different sources/locations.

Figure 1: Synthetic MNIST and Gaussian datasets.

where $2a$ is the number of degrees of freedom, and $L = \frac{a\kappa}{b(\kappa+1)}$. Using these, our predictive function (in log-space) takes the form

$$\arg\min_y \ \frac{\lambda_x}{2}\Big(y - f(x)\Big)^2 - \frac{\lambda_y}{2}\Big(y - m\Big)^2 + \frac{2a+1}{2}\log\Big(1 + \frac{L}{2a}(y - \mu)^2\Big). \qquad (19)$$

Setting the derivative $wrt$ $y$ to zero yields a cubic equation (see Appendix B.1), which we can solve to find roots. A positive sign of the second derivative of the objective tells us if a solution is a (local) minima. When we have one real solution with a positive second derivative, we use this; when we have multiple real solutions with positive second derivatives, we pick the one that corresponds to the smallest objective; when we have no real solutions with positive second derivatives, we do not update $\mathbb{P}(y \mid x)$, retaining $f(x)$ as the solution. Empirically, we find that the condition for no local minima does not arise for optimal choices of hyper-parameters (also see Appendix B.2).

The update equations at the $t$-th step follow from the computation of the posterior using Eq. 13 (see Murphy (2007), for example, for the derivation of these update steps) and are given as:

$$a_{t+1} = a_t + 1/2; \ \ \kappa_{t+1} = \kappa_t + 1; \ \ \mu_{t+1} = \frac{\kappa_t\mu_t + \hat{y}_{t+1}}{\kappa_t + 1}; \ \ b_{t+1} = b_t + \frac{\kappa_t(\hat{y}_{t+1} - \mu_t)^2}{2(\kappa_t + 1)}. \qquad (20)$$

The hyper-parameters $\lambda_x$ (output precision) and $\kappa$ (equivalent of the smoothing pseudo-count $\alpha_0$ in classification) are picked on the validation set, along with a scaling pre-multiplier for the precision $\lambda_y$ (analogous to the classification setup). In order to place uniform priors over the output range, we will simulate a uniform set of samples over the output range. $\mu = \mathbb{E}[y^{\text{pseudo}}]$ is the mean of the pseudo-samples, and $\beta$ is initialized as $0.5(\kappa - 1)\text{Var}(y^{\text{pseudo}})$ (see Appendix B.3 for details).

## 5 EXPERIMENTS

We compare variants of online label-shift methods based on our discussion above on a mix of synthetic and realistic datasets to the un-adjusted model performance (BASE).

- FTH and OGD: These are the variants proposed in Wu et al. (2021). We evaluate both for two choices of confusion matrices each – computed using the in-distribution validation set, and using the out-of-distribution validation set (our HEURISTIC 1). We refer to these two alternatives as (C-IID) and (C-OOD).

- FTH-H and OGD-H: These are our modifications of FTH and OGD using the scaling hyper-parameters proposed in HEURISTIC 2. For both variants, we again evaluate two versions each, using (C-IID) and (C-OOD).

- FTH-H-B: This is our modification of FTH, with an additional pseudo-count hyper-parameter added for smoothing. The hyper-parameters are learned on the OOD validation sets. We call the regression variant FTH-H-B (R).

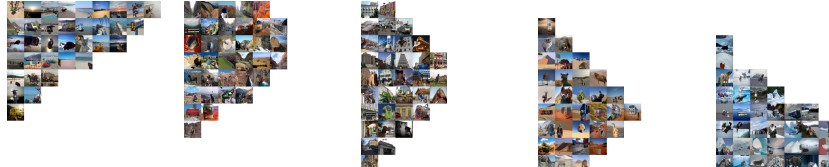

Figure 2: Skewed COCO-on-Places: Synthetic dataset constructed by superimposing COCO objects (Lin et al., 2014) on scenes from the Places dataset (Zhou et al., 2017). The 5 columns correspond to 5 sources of data, where the backgrounds correspond to examples of particular scenes, and the skew in number of examples per row correspond to the skew in label distribution we impose. Different background scenes are used for training, validation, and test sets.

- OPTIMAL FIXED CLASSIFIERS: These oracle methods are derived by replacing **p** in Eq. 2 with the empirical location-wise label distributions, providing a sense of achievable gains if one were aware of the true label-distributions from the get-go. We include two variants – OFC, which uses Eq. 2, and OFC-H, which uses the modified update rule in Eq. 8 where the hyper-parameters are oracle hyper-parameters learned on the test-set.

When using OGD, we use the surrogate loss implementation in Wu et al. (2021) since it is both better-performing as well as much faster. This variant involves using a smooth approximation of the 0-1 loss allowing for direct gradient computation instead of a numerical approximation.

## 5.1 CLASSIFICATION PROBLEMS

### 5.1.1 SYNTHETIC: SKEWED-MNIST

We split MNIST classes into two subsets: [0, 1, 2, 5, 9] and [3, 4, 6, 7, 8]. We use different colors to correspond to different deployment locations, similar to Arjovsky et al. (2019). In the training set, we color digits in a particular subset a particular color 99% of the time. This corresponds to a 99% skew in label-distributions across the two locations. The 1% cross-over instructs some color-invariance but not strongly enough to completely overcome the bias. The validation set uses opposing colours for the subsets, but with a 75% correlation – this represents a scenario where the class-distributions in different locations change from that in training. Finally, the test set uses completely flipped colors in the two subsets compared to the training set – this implies reversed label-distributions, resulting in poorer baseline performance.

Since the overall class frequencies are balanced in the training set, we drop the $P(y)$ from the update rule in Eq. 2 and 8. With a 3-layer CNN trained for 20 epochs to 100% training set accuracy and 99.6% in-distribution test set accuracy, we find, in Table 1, that using online adjustments at test-time can lead to marked improvements for the base model in the test set. The numbers are averaged over 5 independent rounds of base-model training, with validation and test sets randomly shuffled for 5 trials for each round of training. (More details about dataset construction in Appendix C.1)

### 5.1.2 SYNTHETIC: SKEWED-COCO-ON-PLACES

We construct a second, more photo-realistic, synthetic dataset by superimposing segmented objects from COCO Lin et al. (2014) on to scenes from the PLACES dataset Zhou et al. (2017), as in Ahmed et al. (2021). The scenes correspond to the notion of a deployment location, albeit with significant intra-location variation. For every such scene-represented source, we use a different class-distribution to simulate source-specific skews in the label distribution. In Fig. 2 the relative number of images per row represent the relative frequency of a particular class at a specific source. There are a total of $\sim 10K$ training images, $\sim 2.5K$ validation images (each for seen and unseen sources), and $\sim 6K$ test images (each for seen and unseen sources).

The validation and test sets are constructed similarly. For in-distribution validation and test sets, the same set of scenes as for training is used (with different instances), and for new-location validation and test sets, different sets of scenes are used. See Appendix C.3 for details about dataset construction. We train a ResNet-50 for 400 epochs with SGD+Momentum for the underlying model, achieving an in-distribution test accuracy of $\sim 75\%$. Since the overall distribution of classes is close to uniform,

Table 1: Classification problems: Average accuracy on SKEWED-MNIST, SKEWED-COCO-ON-PLACES, and WILDS-IWILDCAM (also reporting macro F1-score for IWILDCAM). Overall trends indicate that our heuristics are helpful, and FTH-H-B is competitive or better without needing a confusion matrix.

| Method | S-MNIST | S-COCO-ON-PLACES | IWILDCAM (Avg.) | IWILDCAM (F1) |
|---|---|---|---|---|
| BASE | $82.59 \pm 1.82$ | $56.09 \pm 0.66$ | $73.10 \pm 3.26$ | $32.70 \pm 0.16$ |
| FTH (C-IID). | $93.12 \pm 1.57$ | $58.50 \pm 0.55$ | $71.41 \pm 4.91$ | $29.57 \pm 0.93$ |
| FTH (C-OOD) | $96.04 \pm 1.03$ | $58.94 \pm 0.63$ | $71.41 \pm 4.91$ | $29.57 \pm 0.93$ |
| OGD (C-IID) | $88.32 \pm 2.06$ | $57.37 \pm 0.51$ | $71.66 \pm 4.56$ | $32.56 \pm 0.27$ |
| OGD (C-OOD) | $95.75 \pm 0.70$ | $57.75 \pm 0.29$ | $73.11 \pm 3.05$ | $32.49 \pm 0.41$ |
| FTH-H (C-IID) | $98.21 \pm 0.47$ | $56.72 \pm 0.84$ | $73.75 \pm 3.77$ | $32.46 \pm 0.31$ |
| FTH-H (C-OOD) | $98.69 \pm 0.31$ | $57.81 \pm 0.74$ | $73.75 \pm 3.77$ | $32.46 \pm 0.31$ |
| OGD-H (C-IID) | $96.07 \pm 1.76$ | $57.58 \pm 0.79$ | $72.89 \pm 3.30$ | $31.74 \pm 0.51$ |
| OGD-H (C-OOD) | $98.91 \pm 0.20$ | $57.12 \pm 0.15$ | $73.36 \pm 3.51$ | $31.36 \pm 0.41$ |
| FTH-H-B | $97.46 \pm 0.64$ | $58.42 \pm 0.49$ | $74.10 \pm 3.56$ | $33.33 \pm 1.31$ |
| OFC | $99.24 \pm 0.20$ | $75.88 \pm 0.33$ | $79.19 \pm 1.76$ | $48.61 \pm 0.27$ |
| OFC-H | $99.26 \pm 0.20$ | $75.88 \pm 0.33$ | $81.07 \pm 0.79$ | $48.61 \pm 0.27$ |

we again drop the marginal $P(y)$ term in Eq. 2 and 8. In Table 1 we again find improved performance over the unadjusted base model for all variants. Accuracy is aggregated across 20 random orderings of the test set (since the test-sets are smaller for this specific dataset), for 3 rounds of base-model training each.

### 5.1.3 WILDS-IWILDCAM

We use the variant of the IWILDCAM 2020 dataset Beery et al. (2021) curated by the WILDS set of benchmarks for out-of-distribution (OOD) generalization Koh et al. (2021). The data consists of burst images taken at camera traps, triggered by animal motion. The task is to identify the species in the picture, and the locations correspond to the unique camera trap the pictures are from. There are a total of 182 species in this version of the dataset across a total of 323 camera traps. There is significant skew in terms of species distribution across different camera traps, as well as the number of images available for each trap. The training set consists of $\sim 130K$ images from 243 traps; the in-distribution validation set consists of $\sim 7.3K$ images from the same traps as that in the training set but on different dates; the OOD validation set consists of $\sim 15K$ images taken at 32 traps that are different from the ones in the training set; the in-distribution test set consists of $\sim 8.1K$ images taken by the same camera traps as in the training set, but on different dates from both training and validation; finally, the OOD test set consists of $\sim 43K$ images taken at 48 camera traps that are different from those for all other splits.

Koh et al. (2021) trained ResNet-50 based models along with their curation of this dataset, also evaluating several methods for OOD generalization and releasing all models. We use their models trained with the domain generalization method CORAL Sun & Saenko (2016), since this model has improved performance over the ERM baseline. They released three sets of weights, trained with three random seeds. We evaluate all variants for each of the three seeds, with 3 random orderings each of the test set, and report aggregates in Table 1. Koh et al. (2021) recommend evaluation with both average accuracy as well as macro-F1 (since some species in the dataset are rare). We perform evaluation with both metrics, but use our own trained models for average accuracy – this is because Koh et al. (2021) trained their models optimizing for macro F1. We similarly trained CORAL-augmented base models optimizing the penalty coefficient and choice of early stopping.

We replace the confusion matrix with an identity matrix for evaluating methods on this dataset (for methods where a validation-set estimated confusion matrix is required). Confusion matrices evaluated on the validation sets are non-invertible for this dataset due to sparse class-representation and we found common alternatives to perform poorly (see Appendix E).

Table 2: Regression problems: For the GAUSSIANS dataset the metric is mean squared error (lower is better), and for the PovertyMap folds the metric is Pearson's correlation co-efficient (higher is better), computed separately for average (ALL) and worst-group (WG) performance.

| | Dataset | BASE | FTH-H-B (R) | | | |
|---|---|---|---|---|---|---|
| | MIX-OF-GAUSSIANS | $9.17 \pm 2.17$ | $4.35 \pm 1.48$ | | | |
| POVERTYMAP Fold | BASE | FTH-H-B (R) | POVERTYMAP Fold | BASE | FTH-H-B (R) | |
| A (ALL) | 0.84 | $0.84 \pm 0.00$ | A (WG) | 0.42 | $0.43 \pm 0.00$ | |
| B (ALL) | 0.83 | $0.82 \pm 0.00$ | B (WG) | 0.52 | $0.50 \pm 0.01$ | |
| C (ALL) | 0.80 | $0.83 \pm 0.00$ | C (WG) | 0.42 | $0.56 \pm 0.01$ | |
| D (ALL) | 0.77 | $0.77 \pm 0.00$ | D (WG) | 0.50 | $0.56 \pm 0.01$ | |
| E (ALL) | 0.75 | $0.75 \pm 0.00$ | E (WG) | 0.34 | $0.37 \pm 0.00$ | |

## 5.2 REGRESSION PROBLEMS

### 5.2.1 SYNTHETIC: MIX-OF-GAUSSIANS

We create a synthetic regression dataset by constructing a curve from a mixture of Gaussians. We pick regions on the $x$-axis to correspond to training, validation, and test sets, such that every set samples data from two regions each, corresponding to two locations (see Appendix C.2). In Figure 1b, we depict the curve, along with sampling indicators for the different sets and sources. The points have been placed at different heights for clearer visualization of overlaps. 500 points are sampled from the two training regions, and 250 each for the validation and test sets from their assigned regions. We train a 3-layer MLP with BatchNorm and ReLU activations and a mean squared loss for 100 epochs, yielding an in-distribution test mean squared error (MSE) of $\sim 0.15$. In Table 2 we find that online updating reduces the OOD test MSE significantly. Results are aggregates over five trials, with a different random sampling of all data, followed by training and validation each time. Full results and more experimental details are in Appendix C.2).

### 5.2.2 WILDS-POVERTYMAP

We use the WILDS variant of a *poverty mapping* dataset Yeh et al. (2020). This is a dataset for estimating average household economic conditions in a region through satellite imagery, measured by an asset wealth index computed from survey data. The data comprises 8-channel satellite images with data from 23 African countries. The locations here correspond to different countries. Due to the smaller size of the dataset, Koh et al. (2021) recommend a five-fold evaluation, where every fold is approximately constructed as follows – 10K images from 13-14 countries in the training set; 1K images from the same countries for in-distribution validation; 1K images from these countries for in-distribution testing; 4K images from 4-5 countries not in the training set for OOD validation; and 4K images from 4-5 countries in neither training nor validation sets for OOD test.

The evaluation metric is Pearson's correlation between predicted economic index vs. actual index, as is standard in the literature (Yeh et al., 2020). Following Koh et al. (2021), we split the assessment into overall average as well as worst-group performance, which picks the worst performance across rural/urban subgroups. As with IWILDCAM, we use the CORAL-augmented base networks and weights released by Koh et al. (2021), but with our retrained versions for average correlation coefficient (since the validation choices for the released weights were for worst group performance). We evaluate separately for each fold (which have quite a bit of variance in base performance) with 5 random orderings of each of the test sets. In Table 2, we find that while there seems generally little to no improvement for average correlation, there are more significant improvements for three of five folds in terms of worst-group performance. As noted in Koh et al. (2021), a wide range of differences along many dimensions such as infrastructure, agriculture, development, cultural aspects play a role not only in determining wealth-distribution, but also in terms of how the features manifest in different places. Such real-world issues imply that validating for OOD performance is bound to be sensitive to problem types and the specific choices of validation sets used to tune hyper-parameters, and the differences that may arise between an OOD validation set and an OOD test set. This issue extends generally to all attempts at OOD generalization.

### 5.3 TAKEAWAYS

Our experiments are generally suggestive of the following takeaways.

- While invertible confusion matrices are not always achievable due to data scarcity (as modelled in our experiments with WILDS-IWILDCAM), a practitioner can adopt confusion-matrix free methods such as FTH-H-B, which we find to provide competitive or improved performance. Using OOD validation sets to estimate confusion matrices can improve results relative to using an IID validation set, although confusion matrices estimated on smaller-sized sets can be noisy.

- Learning additional scaling hyper-parameters can be useful for further improvements. We find this trend to not hold for SKEWED-COCO-ON-PLACES (FTH outperforms FTH-H and FTH-H-B). We suspect this is likely due to instability from the relatively smaller size of the validation set – when picking oracle scaling hyper-parameters on the test set, we achieve an accuracy of $59.37 \pm 0.89$. In Appendix D we compare performance when learning hyper-parameters on different validation sets – IID/OOD/test (oracle).

## 6 RELATED WORK

**Label-shift for classifiers**    Saerens et al. (2002) provides a seminal discussion about adapting the output distribution of a classifier when the test set undergoes label-shift. This approach presumes access to the entire test set up front, or a sufficiently representative sample. More recent works have investigated other ways to estimate label-shift (Lipton et al., 2018; Azizzadenesheli et al., 2019) using confusion matrices, which partially inspired the methods in Wu et al. (2021) that we use as our foundation. It has been recently suggested (Alexandari et al., 2020; Garg et al., 2020) that the simple correction method in Saerens et al. (2002) often outperforms these later methods when combined with calibration. While Alexandari et al. (2020) perform their calibration using a held-out IID validation set for their iterative method, we adapt this strategy to the out-of-distributions setting by picking scaling hyper-parameters on an OOD validation set.

**Test-time training**    Another emerging line of literature focuses on updating neural network parameters using test data without being able to match training statistics with test statistics, due to the potential lack of access to training data for the same topical reasons – data privacy and large datasets. Some examples include updating the Batch-Norm statistics optimizing for minimum test-time entropy Wang et al. (2021), or using self-supervised pseudo-labels to adapt the feature extraction part of the network Liang et al. (2020). Our setup here can be viewed as a form of test-time training, but in a more constrained setting, with inaccessible model parameters and no resources to replicate an onsite-model by querying the black-box model, e.g. using distillation (Hinton et al., 2015).

**Out-of-distribution generalization**    There has been a recent surge in interest for methods aiming to learn stable or invariant features across different domains/environments/groups Sun & Saenko (2016); Arjovsky et al. (2019); Krueger et al. (2020); Sagawa et al. (2020). Such approaches have been demonstrated to be useful for certain types of distributional shifts, such as with improved minority group robustness Sagawa et al. (2020) and systematic generalization Ahmed et al. (2021). Our discussion in this paper is complementary to this set of methods in OOD generalization research. One can use an underlying model trained with cross-group penalties that result in improved OOD generalization, and further improve performance by factoring in useful contextual information.

## 7 CONCLUSION

In this paper, we empirically investigated the effectiveness of online black-box adaptation methods for label-shift when a key underlying assumption of invariant class-conditional input distributions is broken. We found that while existing methods can be effective to an extent regardless of conditional-shift, performance can be improved by adopting intuitive heuristics – in particular, estimating confusion matrices on OOD validation sets, and learning additional scaling hyper-parameters in the output adjustment step to account for shifting distributions.

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

## A  POSTERIOR UPDATE

We derive the posterior update equation (Eq. 13), specifying the conditions under which this rule holds. The key assumption is that in the new deployment location, categories are encountered in an IID manner in the location, i.e., $y_j \perp\!\!\!\perp y_k$.

$$P_t(\phi) = P(\phi \mid y_1, \cdots, y_t), \tag{21}$$

$$= \frac{P(y_1, \cdots, y_t \mid \phi)\, P(\phi)}{\mathbb{P}(y_1, \cdots, y_t)}, \qquad \text{(Bayes rule)} \tag{22}$$

$$\propto P(y_1, \cdots, y_t \mid \phi)\, P(\phi), \qquad \text{(dropping terms independent of } \phi\text{)} \tag{23}$$

$$= \prod_{i=1}^{t} P(y_i \mid \phi)\, P(\phi) \qquad \text{(using assumption } y_j \perp\!\!\!\perp y_k\text{)} \tag{24}$$

$$= P(y_t \mid \phi)\left(\prod_{i=1}^{t-1} P(y_i \mid \phi)\, P(\phi)\right) \qquad \text{(regrouping terms)} \tag{25}$$

$$= P(y_t \mid \phi)\, P_{t-1}(\phi), \qquad \text{(by definition)} \tag{26}$$

## B  REGRESSION MODEL

### B.1  FINDING THE OPTIMAL SOLUTION FROM THE PREDICTIVE RULE

The required distributions are defined as

$$P(y \mid x) \propto \exp\left(-\frac{\lambda_x}{2}\Big(y - f(x)\Big)^2\right), \tag{27}$$

$$P^{\text{new}}(y) \propto \left(1 + \frac{L}{2a}(y - \mu)^2\right)^{-\frac{2a+1}{2}}, \tag{28}$$

$$P(y) \propto \exp\left(-\frac{\lambda_y}{2}\Big(y - m\Big)^2\right), \tag{29}$$

$$\tag{30}$$

which gives us the objective $J = -\log P(y \mid x)$ expressed as

$$J = -\log P(y \mid x) - \log P^{\text{new}}(y) + \log P(y) \tag{31}$$

$$= \frac{\lambda_x}{2}\Big(y - f(x)\Big)^2 - \frac{\lambda_y}{2}\Big(y - m\Big)^2 + \frac{2a+1}{2}\log\left(1 + \frac{L}{2a}(y - \mu)^2\right) \tag{32}$$

The derivative of this objective $wrt$ y is

$$\frac{\partial J}{\partial y} = \lambda_x(y - f(x)) - \lambda_y(y - m) + \frac{\frac{2a+1}{\cancel{2}}\frac{L}{2a}.\cancel{2}.(y - \mu)}{1 + \frac{L}{2a}(y - \mu)^2} \tag{33}$$

$$= \lambda_x(y - f(x)) - \lambda_y(y - m) + \frac{(2a+1)\frac{L}{2a}(y - \mu)}{1 + \frac{L}{2a}(y - \mu)^2} \tag{34}$$

$$= \underbrace{\big(\lambda_x - \lambda_y\big)}_{\tau_d} y + \underbrace{\big(\lambda_y m - \lambda_x f(x)\big)}_{\tau_\mu} + \frac{\overbrace{(2a+1)}^{A}\,\overbrace{\frac{L}{2a}}^{M}(y - \mu)}{1 + \frac{L}{2a}(y - \mu)^2} \tag{35}$$

$$= \tau_d y + \tau_\mu + \frac{AM(y - \mu)}{1 + M(y - \mu)^2} \tag{36}$$

Setting to zero, we have

$$\Big(\tau_d y + \tau_\mu\Big)\Big(1 + M(y - \mu)^2\Big) + AM(y - \mu) = 0 \tag{37}$$

$$\implies \left(\tau_d y + \tau_\mu\right)\left(1 + My^2 + M\mu^2 - 2M\mu y\right) + AM(y - \mu) = 0 \tag{38}$$

$$\implies \tau_d y + M\tau_d y^3 + M\mu^2\tau_d y - 2M\mu\tau_d y^2 + \tau_\mu + M\tau_\mu y^2 + M\tau_\mu\mu^2 - 2M\mu\tau_\mu y + AMy - AM\mu = 0 \tag{39}$$

$$\implies M\tau_d y^3 + (M\tau_\mu - 2M\mu\tau_d)y^2 + (\tau_d + M\mu^2\tau_d - 2M\mu\tau_\mu + AM)y + (\tau_\mu + M\tau_\mu\mu^2 - AM\mu) = 0 \tag{40}$$

which is the equation we shall solve for $y$. We use NUMPY's polynomial solver to find roots. A cubic equation either has one real and a pair of conjugate imaginary roots, or all real roots. We test the real solutions for a positive curvature (implying local minima), and pick the minima resulting in smallest value of the objective $J$.

## B.2 SECOND DERIVATIVE TEST FOR SOLUTIONS

The second derivative of $J$ is given by

$$\tau_d - \frac{2AM^2(y - \mu)^2}{(1 + M(y - \mu)^2)^2} + \frac{AM}{1 + M(y - \mu)^2} \tag{41}$$

Writing $y - \mu$ as $D$, we have

$$\tau_d + \frac{AM}{(1 + MD^2)} - \frac{2AM^2D^2}{(1 + MD^2)^2} = \tau_d + \frac{AM}{1 + MD^2}\left(1 - \frac{2MD^2}{1 + MD^2}\right) = \tau_d + \frac{AM(1 - MD^2)}{(1 + MD^2)^2} \tag{42}$$

When this expression is positive, we have a local minima.

For the first term to be positive, we require that $\tau_d > 0$, which has a straightforward intuitive interpretation: $\tau_x > \tau_y$, i.e. output precision should be higher than marginal-adjustment precision. This is a reasonable condition which we expect to be fulfilled, since we typically expect to rely more strongly on the underlying predictive model than simply the marginal.

In the second term, $AM$ is always non-negative, for a positive pseudo-count. The denominator is always positive. Substituting in expressions for the values after the $t$-th update, we have

$$MD^2 = \frac{\frac{\kappa_t}{\kappa_t + 1}(y - \mu_t)^2}{\sum_{\tau=0}^{t-1}\frac{\kappa_\tau}{\kappa_\tau + 1}(\hat{y}_{\tau+1} - \mu_\tau)^2}. \tag{43}$$

When this term is $\leq 1$, we are guaranteed positivity (strictly speaking, $\tau_d$ provides the second term with some room for negative values, but we ignore it for simplified reasoning). This condition implies

$$(y - \mu_t)^2 \leq \frac{\kappa_t + 1}{\kappa_t}\sum_{\tau=0}^{t-1}\frac{\kappa_\tau}{\kappa_\tau + 1}(\hat{y}_{\tau+1} - \mu_\tau)^2, \tag{44}$$

which then implies that the following range for $y$ allows local minima

$$\mu_t - \sqrt{\frac{\kappa_t + 1}{\kappa_t}\sum_{\tau=0}^{t-1}\frac{\kappa_\tau}{\kappa_\tau + 1}(\hat{y}_{\tau+1} - \mu_\tau)^2} \leq y \leq \mu_t + \sqrt{\frac{\kappa_t + 1}{\kappa_t}\sum_{\tau=0}^{t-1}\frac{\kappa_\tau}{\kappa_\tau + 1}(\hat{y}_{\tau+1} - \mu_\tau)^2}. \tag{45}$$

An intuitive interpretation of this condition is that valid updates are allowed within an increasing range as a function of the total observed variances up to the $t$-th test example. In practice, we find that validation tends to pick values for $\tau_x > \tau_y$, and that the case for no-local-minima typically does not arise for the optimal hyper-parameters in our experiments.

## B.3 INITIALIZING PRIORS

For initializing priors, we might endeavour to stay unbiased, since we assume that deployment locations can have significantly different target distributions than we might anticipate from the marginal over the training set. For classification, we built this in by using a uniform pseudo-count for all classes and sources. For regression, we simulate a pseudo-count of uniform samples from the output range.

If we start with a reference prior for the Normal-Gamma distribution with parameter settings

$$\mu = ., \kappa = 0, \alpha = -0.5, \beta = 0, \tag{46}$$

then after observing a $N$ data-points $\{y_1, \cdots, y_N\}, y_i \sim \mathcal{U}[L, H]$ (the uniformly sampled points we will simulate), the resulting posterior is

$$\mu = \frac{1}{N} \sum_{i=1}^{N} y_i, \tag{47}$$

$$\kappa = N, \tag{48}$$

$$\alpha = \frac{N-1}{2}, \tag{49}$$

$$\beta = \frac{1}{2} \sum_{i=1}^{N} (y_i - \mu)^2. \tag{50}$$

In this view, $\kappa$ corresponds to the pseudo-count (as per the interpretation of the parameters of the Normal-Gamma conjugate prior as in Murphy (2007)). $\alpha$ is defined in terms of $\kappa$. To improve stability, we will set $\mu$ to the middle of the output range rather than actually estimate the mean of our uniform pseudo-samples. Likewise, we will set $\beta$ by estimating its value as a function of $\kappa$ and using the expression for variance of a uniform distribution,

$$\mathbb{E}[\beta] = \frac{1}{2}(\kappa - 1)\text{Var}(y_i) = (\kappa - 1)\frac{(H-L)^2}{24}. \tag{51}$$

## C  EXPERIMENTAL DETAILS

### C.1  SYNTHETIC MNIST

The splitting of digits into two sets is performed by observing mis-classification matrices after 200 iterations of training a neural network averaged across a 100 runs – digits are put into opposing sets if they tend to be confused, while also trying to keep the set-sizes balanced.

The network architecture consists of 3 CONV layers with 64, 128 and 256 channels, each followed by MAXPOOL, BATCHNORM, and RELU. After the third layer, we spatially mean-pool activations and use a linear layer to map to the logits. A weight-decay of $5e - 4$ is applied on all parameters. Training is conducted for 20 epochs with batches of size 256 where training accuracy saturates to 100%. An initial learning rate of 0.1 is used, which is cut by 5 at the 6-th, 12-th and 16-th epochs.

The datapoint-counts in the train/val/test environments are as follows.

|  |  | 0 | 1 | 2 | 3 | 4 | 5 | 6 | 7 | 8 | 9 |
|---|---|---|---|---|---|---|---|---|---|---|---|
| Train | red | 4889 | 5614 | 4915 | 38 | 49 | 4664 | 57 | 59 | 41 | 4946 |
|  | cyan | 43 | 64 | 53 | 5063 | 4810 | 42 | 4894 | 5116 | 4801 | 42 |
| IID validation | red | 989 | 1052 | 985 | 7 | 10 | 904 | 12 | 8 | 12 | 949 |
|  | cyan | 2 | 12 | 5 | 1023 | 973 | 11 | 955 | 1082 | 997 | 12 |
| OOD validation | cyan | 687 | 714 | 689 | 313 | 304 | 635 | 310 | 315 | 301 | 664 |
|  | red | 304 | 350 | 301 | 717 | 679 | 280 | 657 | 775 | 708 | 297 |
| OOD Test | cyan | 980 | 1135 | 1032 | 0 | 0 | 892 | 0 | 0 | 0 | 1009 |
|  | red | 0 | 0 | 0 | 1010 | 982 | 0 | 958 | 1028 | 974 | 0 |

### C.2  SYNTHETIC GAUSSIAN

The synthetic data for this experiment is generated with the following function

$$y(x) = 10\mathcal{N}(y \mid x; \mu = -2, \sigma = 0.5) + 3\mathcal{N}(y \mid x; \mu = 2, \sigma = 0.5) + 6\mathcal{N}(y \mid x; \mu = 0, \sigma = 1)$$

**Training points:** Training points are sampled from two regions on the $x$-axis, $x \sim \mathcal{N}(-2, 0.4)$ and $x \sim \mathcal{N}(2, 0.2)$, with 250 points each.

**OOD validation points:** OOD validation points are sampled from $\mathcal{N}(-3.5, 0.2)$ and $\mathcal{N}(1, 0.2)$, with 250 points each.

**OOD test points:** OOD test points are sampled from $\mathcal{N}(0, 0.2)$ and $\mathcal{N}(3, 0.2)$, with 250 points each.

For OOD sets, the different sampling distributions correspond to different locations. For different trials, we repeat the whole experiment from scratch, sampling new training, validation, and test sets, and performing validation every time.

The network architecture is a 3 layer MLP with 128 hidden units, with BATCHNORM and RELU after hidden activations. A weight decay of $1e - 8$ is applied on all parameters. We train for a 100 epochs with batch-sizes of 100, with SGD + Momentum (0.9), starting with an initial learning rate of 0.01 and scaling it by 0.95 after every epoch.

We include the non-aggregated MSEs below to confirm that there are consistent improvements over every base model/data-sampling individually.

| Seed | IID-Base | OOD-Base | OOD-Online |
|---|---|---|---|
| 0 | 0.08 | 11.23 | 3.14 |
| 1 | 0.13 | 12.37 | 3.82 |
| 2 | 0.16 | 6.13 | 3.00 |
| 3 | 0.19 | 9.14 | 5.50 |
| 4 | 0.21 | 7.00 | 6.31 |

### C.3 Synthetic Skewed-COCO-on-Places

We chose the following objects for this synthetic classification task: *bicycle*, *train*, *cat*, *chair*, *horse*, *motorcycle*, *bus*, *dog*, *couch*, and *zebra*; and the following scenes to simulate different sources.

**Training**: beach, canyon, building_facade, desert/sand, iceberg

**OOD validation:** oast_house, orchard, crevasse, ball_pit, viaduct

**OOD test:** water_tower, staircase, waterfall, bamboo_forest, zen_garden

When there are multiple instances of a class in an image, we pick the instance occupying largest area, such that only images with objects occupying at least 10K pixels are retained. All images are resized to $256 \times 256$.

Across the 5 sources, the number of examples for training, validation, and test sets are as follows.

Table 3: Training set

| | bicycle | train | cat | chair | horse | motorcycle | bus | dog | couch | zebra |
|---|---|---|---|---|---|---|---|---|---|---|
| beach | 669 | 669 | 429 | 176 | 46 | 7 | 0 | 0 | 0 | 0 |
| canyon | 135 | 329 | 513 | 513 | 329 | 135 | 35 | 6 | 0 | 0 |
| building_facade | 5 | 34 | 132 | 322 | 503 | 503 | 322 | 132 | 34 | 5 |
| desert/sand | 0 | 0 | 6 | 35 | 135 | 329 | 513 | 513 | 329 | 135 |
| iceberg | 0 | 0 | 0 | 0 | 7 | 46 | 176 | 429 | 669 | 669 |

Table 4: Validation sets

| | bicycle | train | cat | chair | horse | motorcycle | bus | dog | couch | zebra |
|---|---|---|---|---|---|---|---|---|---|---|
| beach | 167 | 167 | 107 | 44 | 11 | 1 | 0 | 0 | 0 | 0 |
| canyon | 33 | 82 | 128 | 128 | 82 | 33 | 8 | 1 | 0 | 0 |
| building_facade | 1 | 8 | 33 | 80 | 125 | 125 | 80 | 33 | 8 | 1 |
| desert/sand | 0 | 0 | 1 | 8 | 33 | 82 | 128 | 128 | 82 | 33 |
| iceberg | 0 | 0 | 0 | 0 | 1 | 11 | 44 | 107 | 167 | 167 |

Table 5: Test sets

|  | bicycle | train | cat | chair | horse | motorcycle | bus | dog | couch | zebra |
|---|---|---|---|---|---|---|---|---|---|---|
| beach | 401 | 401 | 257 | 105 | 27 | 4 | 0 | 0 | 0 | 0 |
| canyon | 81 | 197 | 308 | 308 | 197 | 81 | 21 | 3 | 0 | 0 |
| building_facade | 3 | 20 | 79 | 193 | 302 | 302 | 193 | 79 | 20 | 3 |
| desert/sand | 0 | 0 | 3 | 21 | 81 | 197 | 308 | 308 | 197 | 81 |
| iceberg | 0 | 0 | 0 | 0 | 4 | 27 | 105 | 257 | 401 | 401 |

Note that the pattern of label-shift is the same across validation and test subsets (albeit of a smaller size). This proof-of-concept experiment is intended as a middle-ground between the COLORED MNIST and WILDS-IWILDCAM experiments, in that the potential of learning hyper-parameters to account for conditional shift is tested while keeping label-shift pattern fixed).

We train for 400 epochs with SGD + Momentum (0.9), using batch sizes of 128, with an initial learning rate of 0.1 which is cut by 5 at the 240th, 320th, 360th epochs. An L2 weight decay regulariser is applied on all parameters with a coefficient of $5e-4$. We normalize images with the training set mean and standard deviation per channel, and apply data augmentation of random crops to $224 \times 224$ and random horizontal reflections.

## D   HYPER-PARAMETER SELECTION

We contrast performance when methods use IID validation sets vs. OOD validation sets vs. the test set itself, in Table 6 . We observe that, generally speaking, OOD validation can improve over IID validation.

## E   IDENTITY APPROXIMATION FOR CONFUSION MATRIX

Degenerate confusion matrices can arise when there are missing categories in the validation set used to compute it (leading to zero-rows), or if two or more rows are exactly the same (for example, when multiple rare categories both get categorized the same way). Two options are to use a soft-confusion matrix, or a pseudo-inverse (Lipton et al., 2018). Since the IWILDCAM dataset is significantly long-tailed, with a large number of classes not represented in the validation sets, we end up with a number of zero rows for the soft-confusion matrix. For such rows, we simply placed a 1 in the diagonal element.

In Table 7, we find these alternatives to result in degraded performance for IWILDCAM, generally much worse than our identity approximation. We hypothesize that part of the reason is to do with the fact that both our zero-confusion heuristic for dealing with missing classes for the soft-confusion matrix, as well as the same underlying effect being applied by the pseudo-inverse results in a misleading effect: rare classes, absent from validation sets, are in fact more likely to be confused than the frequent ones. This is one possibility for why the less presumptive identity approximation performs better. The inherent difficulty is estimating robust confusion matrices has been recognized in the literature, with the typical approach being to hold out significantly large validation sets in order to reliably estimate less noisy confusion matrices. In Table 8, we include numbers from an identity approximation in the synthetic datasets where the confusion matrices were invertible.

On the whole, we suggest to practitioners that in difficult, real-life situations, simpler approximations might continue to serve us well, while more sophisticated methods can pose specific requirements to be successful.

## F   HYPERPARAMETERS, COMPUTE, AND CODE AND DATA LICENSES.

The hyper-parameters involved are the two calibration terms $\lambda_u, \lambda_y$ and the pseudo-count term $\alpha_0$ for classification, and $\lambda_x, \lambda_y, \kappa$ for the regression problems. These were picked via grid-search on the OOD validation sets, optimizing for OOD performance in all cases. For OGD methods, an additional

Table 6: (top) Classification problems: Performance when picking hyper-parameters on IID, OOD validation sets, or on (Oracle) test sets. (bottom) Regression problems: Performance when picking hyper-parameters on IID, OOD validation sets, or on (Oracle) test sets. For MIX-OF-GAUSSIANS, we use mean squared error as the metric (lower is better), while for POVERTYMAP the metric is the Pearson's correlation co-efficient (higher is better).

| Datasets | Methods | IID validation | OOD validation | Oracle |
|---|---|---|---|---|
| S-MNIST | FTH-H | $82.67 \pm 1.79$ | $98.69 \pm 0.30$ | $98.69 \pm 0.30$ |
| | OGD | $82.75 \pm 1.77$ | $95.75 \pm 0.70$ | $95.75 \pm 0.70$ |
| | OGD-H | $82.59 \pm 1.82$ | $98.91 \pm 0.20$ | $98.91 \pm 0.20$ |
| | FTH-H-B | $83.00 \pm 1.79$ | $97.46 \pm 0.64$ | $98.35 \pm 0.52$ |
| S-COCO-ON-PLACES | FTH-H | $57.42 \pm 0.53$ | $57.81 \pm 0.74$ | $59.05 \pm 0.53$ |
| | OGD | $57.72 \pm 0.31$ | $57.75 \pm 0.29$ | $57.75 \pm 0.29$ |
| | OGD-H | $57.31 \pm 0.68$ | $57.12 \pm 0.15$ | $58.10 \pm 0.85$ |
| | FTH-H-B | $58.59 \pm 1.02$ | $58.42 \pm 0.49$ | $59.37 \pm 0.89$ |
| IWILDCAM (AVG) | FTH-H | $73.52 \pm 3.36$ | $73.75 \pm 3.77$ | $74.13 \pm 3.54$ |
| | OGD | $69.42 \pm 5.10$ | $73.11 \pm 3.05$ | $73.16 \pm 3.15$ |
| | OGD-H | $73.41 \pm 3.42$ | $73.36 \pm 3.51$ | $73.53 \pm 3.29$ |
| | FTH-H-B | $73.90 \pm 3.93$ | $74.10 \pm 3.56$ | $74.41 \pm 3.65$ |
| IWILDCAM (F1) | FTH-H | $31.93 \pm 1.56$ | $32.46 \pm 0.31$ | $33.81 \pm 0.30$ |
| | OGD | $29.37 \pm 2.15$ | $32.49 \pm 0.41$ | $32.72 \pm 0.06$ |
| | OGD-H | $32.09 \pm 0.29$ | $31.36 \pm 0.41$ | $32.72 \pm 0.15$ |
| | FTH-H-B | $32.73 \pm 2.78$ | $33.33 \pm 1.31$ | $33.33 \pm 1.31$ |

| Datasets | IID validation | OOD validation | Oracle validation |
|---|---|---|---|
| MIX-OF-GAUSSIANS | $9.24 \pm 2.76$ | $4.35 \pm 1.48$ | $1.76 \pm 0.59$ |
| POVERTYMAP-A (ALL) | $0.80 \pm 0.00$ | $0.84 \pm 0.00$ | $0.84 \pm 0.00$ |
| POVERTYMAP-B (ALL) | $0.82 \pm 0.00$ | $0.82 \pm 0.00$ | $0.83 \pm 0.00$ |
| POVERTYMAP-B (ALL) | $0.82 \pm 0.00$ | $0.83 \pm 0.00$ | $0.83 \pm 0.00$ |
| POVERTYMAP-B (ALL) | $0.78 \pm 0.01$ | $0.77 \pm 0.00$ | $0.78 \pm 0.00$ |
| POVERTYMAP-B (ALL) | $0.72 \pm 0.01$ | $0.75 \pm 0.00$ | $0.75 \pm 0.00$ |
| POVERTYMAP-A (WG) | $0.43 \pm 0.00$ | $0.43 \pm 0.00$ | $0.45 \pm 0.02$ |
| POVERTYMAP-A (WG) | $0.33 \pm 0.03$ | $0.50 \pm 0.01$ | $0.52 \pm 0.00$ |
| POVERTYMAP-A (WG) | $0.50 \pm 0.01$ | $0.56 \pm 0.01$ | $0.58 \pm 0.02$ |
| POVERTYMAP-A (WG) | $0.46 \pm 0.04$ | $0.56 \pm 0.01$ | $0.57 \pm 0.02$ |
| POVERTYMAP-A (WG) | $0.36 \pm 0.02$ | $0.37 \pm 0.00$ | $0.37 \pm 0.00$ |

hyper-parameter is the learning rate used for updating $\mathbf{p}$. This learning rate is searched over a range from $1e$-8 to 10 in steps of $\times 10$.

V100 GPUs were used to train base models (in cases where we trained our own models), and the online adjustment experiments were performed on an Apple Macbook Air with saved outputs from the models.

We reused code from `https://github.com/p-lambda/wilds`, released under the MIT License, and code from `https://github.com/wrh14/online_adaption_to_label_distribution_shift`, publicly released by Wu et al. (2021). We also used data from MS-COCO, released under the CREATIVE COMMONS ATTRIBUTION 4.0 LICENSE. WILDS-IWILDCAM is under COMMUNITY DATA LICENSE AGREEMENT – PERMISSIVE – V1.0, and the WILDS-POVERTYMAP data is U.S. PUBLIC DOMAIN (LANDSAT/DMSP/VIIRS).

Table 7: We compare use of a soft-confusion matrix and the pseudo-inverse with our approximation with an identity matrix for IWILDCAM. We find that FTH performance drops strongly, and for OGD, the optimal learning rate is most often zero, leading to no differences with base performance. For OGD, we find the optimal learning rate on the test-set for all choices of confusion matrix, reporting best-case performance.

| Dataset | Method | Soft confusion matrix | Pseudo-Inverse | Identity |
|---|---|---|---|---|
| IWILDCAM (AVG) | FTH (C-IID) | $43.41 \pm 21.80$ | $37.23 \pm 19.34$ | $71.41 \pm 4.91$ |
| | FTH (C-OOD) | $34.56 \pm 16.71$ | $28.20 \pm 13.74$ | $71.41 \pm 4.91$ |
| | OGD (C-IID) | $73.10 \pm 3.26$ | $73.29 \pm 3.04$ | $73.16 \pm 3.33$ |
| | OGD (C-OOD) | $73.10 \pm 3.26$ | $73.10 \pm 3.26$ | $73.17 \pm 3.18$ |
| IWILDCAM (MACRO-F1) | FTH (C-IID) | $22.42 \pm 4.33$ | $11.33 \pm 0.26$ | $29.57 \pm 0.93$ |
| | FTH (C-OOD) | $23.73 \pm 3.36$ | $10.82 \pm 4.64$ | $29.57 \pm 0.93$ |
| | OGD (C-IID) | $32.71 \pm 0.18$ | $32.70 \pm 0.16$ | $32.75 \pm 0.17$ |
| | OGD (C-OOD) | $32.71 \pm 0.14$ | $32.70 \pm 0.16$ | $32.70 \pm 0.16$ |

Table 8: Identity approximation with S-MNIST and S-COCO-ON-PLACES, with test-time performance using the original confusion matrix $C_f$ for reference. When using the identity approximation, OGD (IID) uses the IID validation set to estimate $C_g$ and OGD (OOD) uses the OOD validation set.

| Dataset | Method | Identity approximation | Original |
|---|---|---|---|
| S-MNIST | FTH | $96.02 \pm 1.07$ | $96.04 \pm 1.03$ |
| | OGD (IID) | $89.47 \pm 1.96$ | $88.32 \pm 2.06$ |
| | OGD (OOD) | $95.70 \pm 0.68$ | $95.75 \pm 0.70$ |
| S-COCO-ON-PLACES | FTH | $59.27 \pm 0.64$ | $58.94 \pm 0.63$ |
| | OGD (IID) | $57.48 \pm 0.52$ | $57.37 \pm 0.51$ |
| | OGD (OOD) | $56.02 \pm 0.35$ | $57.75 \pm 0.29$ |

