# OpenReview forum: "Online black-box adaptation to label-shift in the presence of conditional-shift"
_ICLR.cc/2023/Conference — Submitted to ICLR 2023_

### Official Review · Reviewer_iFod · 2022-10-21

**Confidence:** 4
**Correctness:** 2
**Technical Novelty And Significance:** 1
**Empirical Novelty And Significance:** 1
**Recommendation:** 3

**Clarity, Quality, Novelty And Reproducibility:**

Clarity:
The paper does not seem to be self contained and some necessary details seem to be explained at a very high level. this makes it very hard to understand the methodology, motivation etc.

Novelty and QUality:
limited novelty and technical contribution - essentially the heuristics

Reproducibility:
Because of the high level intuitions, and lack of details, one may not be able to reproduce the results.

**Strength And Weaknesses:**

Weakness:
1. The writeup does not seem to be self contained and the reader may have to refer to Wu et.al.'21 to understand some background.
2. Some parts may require more details. For example, how is p estimated in (8) ? Do the additional hyperparameters effect p's estimation?
3. While the Bayesian model equations are standard, the connection of it for the problem at hand in section 4 is not clear to me. Perhaps some re-writing might help here.
4. Apart for some cryptic justification, none of the propositions seem to have theoretical justifications. This makes it very hard for me to evaluate the work.

**Summary Of The Paper:**

Using standard label shift handling techniques/equations, it is proposed to handle additional manifestation-shift (p(x/y) shift) by simple heuristics based on additional hyperparameters (like in (8)) or using Bayesian models over labels etc. (19). Empirically it is shown that the proposed heuristics improve over those in Wu et.al.'21.


**Summary Of The Review:**

This seems to be a work in preliminary form with many missing details and justifications. Hence I tend to not accept the paper.

---

> ### Author Response · Authors · 2022-11-19
> **Response to review**
>
> Thanks for the review!
>
> > 1. Writing not self-contained; reader might have to consult background literature
>
> We apologize for the lack of self-containedness. We have tried to emphasize some of the key missing background information suggested by the reviews.
>
> > 2. Estimation of p
>
> p is estimated by the OGD and FTH methods described in Section 2.1. The additional hyper-parameters do not influence the FTH estimation of p, but they can technically be included in OGD’s estimation of the class-wise error-rate. However, in early experiments when we attempted including the revised learning rule within OGD’s p-estimation step, we found it to hurt performance. We hypothesize that this might be because our hyper-parameters are learned in an online setting on a specific validation set, while OGD’s estimation of the class-wise error rate is performed in aggregate (on potentially different validation set, as in our C-IID experiments). While more refined alternatives might be possible, in practice, our heuristic for now is to only apply the additional hyper-parameters at the predictive step, with p estimated as in Wu et al.
>
> > 3. Relevance of the Bayesian connection
>
> The Bayesian interpretation of FTH leads us to the FTH-H-B method for classification problems, and the generality of the update equations helps us easily derive analogous updates for regression. The model follows from Section 3.2, where we approximate FTH by using an identity confusion matrix, and observe in Section 4 that using a smoothing vector is equivalent to the update equations 13 and 14. We have rewritten parts of 3.2 to make the transition clearer.
>
> > 4. Lack of theoretical justification
>
> We agree that we do not provide any theoretical justifications behind our proposed heuristics. We view our work as more of an empirical study at this point, but a more principled view would definitely be an improvement.

---

### Official Review · Reviewer_s3Qi · 2022-10-23

**Confidence:** 4
**Correctness:** 2
**Technical Novelty And Significance:** 2
**Empirical Novelty And Significance:** 2
**Recommendation:** 3

**Clarity, Quality, Novelty And Reproducibility:**

Quality: the paper lacks a comparison with the baseline, which learns directly with the validation dataset, and the proposed heuristics are somewhat unconvincing. (Please see the second point of the weaknesses for more details.)

Clarity: This paper is well-structured, but some notations are abused, which makes the background part hard to follow. Besides, the difference between this work and the previous one [Wu et al., 2021] on the problem setup is not clearly discussed. It is unclear to me how the validation dataset is collected, which plays an important role in the algorithms design of this paper.

Originality: this paper is an extension of Wu et al., [2020]. Although some heuristics are proposed to improve the previous work, some of them are less convincing to me.

Reproducibility: codes for the experiments are not provided.


**Strength And Weaknesses:**

### strength
+ This paper considers an interesting problem on how to relax the label shift assumption made in the previous work.
+ This paper extends the online label shift problem to the regression setting.
### weaknesses:
- About the written quality: Although this paper is well-organized, some parts of the presentations, particularly the algorithm design part, are not totally clear. The unclear parts are listed as follows.
	- about the background: the notation $f$ in Eq. (2) and Eq. (4) are very confusing. In Eq. (2), $f$ is used for an underlying base model, but Eq. (4)  uses $f$ for the model to be learned.
	- about Heuristic 1: it is unclear to me how the validation set is collected. Does the learner collect the validation data once at the beginning of the testing online test stage, or do the data just appear in an online fashion? The former one seems less promising in the online adaption problem since the underlying distribution $P_t$ could be different for every iteration.
	- about the Bayesian methods: the notation $\hat{y}_\tau$ in Eq. (9) and $y_t$ in Eq.(13) is not defined, though I can guess they are the pseudo-label and the true label. It is unclear to me why Eq. (13) updates with the true label while Eq. (9) updates with the pseudo-label. It is a very strong requirement to obtain the true label for each iteration.

- About the soundness of the proposed heuristics:
	- about the validation set: the main difference between this work and Wu et al., 2020 is that the latter does not require the validation set. When a validate set sampled from the testing distribution is available, a strong baseline is that we can just estimate $\hat{\mathbf{q}}^{\mathrm{new}}$ by the labeled validate set. I think it would be necessary to compare the proposed method with such a baseline.
	- about Heuristics 1: Heuristics 1 aims to solve the problem when the label-shift condition is broken. But the proposed methods are still based on the reweighted classifier Eq (2). Such a kind of reweighting mechanism still crucially relies on the label shift assumption. In this sense, I believe only the adjustment on the estimation of the confusion matrix is not sufficient.
	- about Heuristic 3: it seems that the use of an identity matrix instead of the confusion matrix will sacrifice the unbiasedness of the gradient estimator. I am not sure whether such a method can perform well when the confusion matrix is invertible.


**Summary Of The Paper:**

This paper studies how to adapt a black box to the testing distribution in an online fashion under the label shift condition. The main contributions are to propose several heuristics to improve the algorithm proposed by Wu et al., (2021) when the label-shift assumption is broken, or the confusion matrix is non-invertible. Empirical studies are conducted to validate the effectiveness of the proposed methods.


**Summary Of The Review:**

This paper considers how to improve the previous work [Wu, et al., 2021] to learn beyond the label shift assumption. This is an interesting problem, but the proposed method is somewhat unconvincing to me as it requires an additional validation dataset and the reweighed classifier still implicitly relies on the label shift assumption. Besides, the background, problem setup, and method parts of the paper are not clearly written. Given the above concerns, I tend to reject this paper.

---

> ### Author Response · Authors · 2022-11-19
> **Response to review**
>
> Thanks for the review!
>
> > Writing:
>
>  – We apologize for the confusions; we inherit all of our Background discussion from past work, where f is treated generally as a classifier at first, but then, due to the output-adjustment approach, the classifier f gets eventually replaced by the classifier g in Eq. 5, with the learnable parameters being p and not those of f anymore, which remains a black-box classifier.
>
> –  The validation set is collected at the beginning, but the processing is sequential when selecting hyper-parameters. For the purposes of validation, we do not think it is unrealistic to collect an entire set up front for development, as long as the test-set is strictly sequential. Also, we do not consider any temporal label-drift in our problems, only that due to different deployment locations, where the models are deployed separately.
>
>  – We apologize for not being clear; in practice, since we do not have access to the true label, we do in fact use the pseudo-label in Eq. 13 as well. We have updated the draft to make this clearer.
>
> > Soundness:
>
>  – We agree that the required access to an additional validation set can be a weakness. However, we note that Wu et al. also requires a large hold-out validation set, both to estimate the confusion matrix as well as to estimate the gradient for OGD. We agree with a baseline where we are aware of the true distribution, and have added two such oracle rows to Table 1, OFC and OFC-H.
>
>  – We agree that we are still approximating the predictive rule based on the label-shift assumption. Our take is that as long as an approximation works in practice, it’s worth reporting.
>
>  – When the confusion matrix is invertible, an identity approximation ought to under-perform (as long as the matrix is estimated to a sufficiently high degree of robustness, which is why the literature always involves holding out a significantly large sized portion of data to estimate this). Our point is that such robust estimation might be difficult to achieve in practice when working with long-tailed problems in post-hoc settings.

---

### Official Review · Reviewer_eNZr · 2022-10-23

**Confidence:** 4
**Correctness:** 3
**Technical Novelty And Significance:** 2
**Empirical Novelty And Significance:** 2
**Recommendation:** 3

**Clarity, Quality, Novelty And Reproducibility:**

Clarity and Quality:
It is easy to follow this draft. The proposed idea is easy to understand.

Novelty:
The novelty of the proposed method is limited. There is a lack of discussion of the intuition behind the proposed heuristics. In the related works, heuristic 3 is a common practice.

Reproducibility:
The proposed heuristic is not difficult to implement. The reproducibility is acceptable.

**Strength And Weaknesses:**

Strength:
+ This work studies a well-motivated problem, which covers many real-world applications.

Weakness:
- The authors introduce three heuristics to improve the empirical performance of the algorithm proposed by Wu et al. (2021) when an additional conditional shift appears. There are, however, no significant or distinguishable improvements over the original method based on their empirical studies. For example, in Table 1, the proposed algorithm's performance on S-COCO-ON-PLACES and IWILDCAM (Avg.) is nearly the same compared with the original algorithm, and even worse than the original one in IWILDCAM (F1).
- Heuristic 3 (adding a tunable scalar to the diagonal and renormalizing rows to the confusion matrix) is a common practice to avoid the non-invertible problem.
- A discussion of the intuition behind the heuristics is suggested.

**Summary Of The Paper:**

This draft considers the problem of online learning with label shift in the presence of additional conditional shift. In addition to the change in the class-priors $\Pr[y]$, the posterior probability $\Pr[x|y]$ can also change over time. Based on the previous work of Wu et al. (2021), the authors propose three heuristics to improve its empirical performance for both classification and regression tasks when the additional conditional shift appears. In their empirical studies, the authors suggest 1) using the OOD validation set instead of the ID validation set when estimating confusion matrices (a key component in the work of Wu et al. (2021)), and 2) adding scaling hyperparameters to the original loss function to improve the performance.

**Summary Of The Review:**

This draft considers the problem of online learning with label shift. The authors propose three heuristics to handle the additional conditional shift based on the previous work of Wu et al. (2021). However, the empirical studies do not show significant improvements over the original approach. The proposed approach lacks theoretical support, and its proposed heuristics lack intuitive support, as well.

---

> ### Author Response · Authors · 2022-11-19
> **Response to review**
>
> Thanks for the review!
>
>  – We agree that the improvements from the modifications and new methods we discussed does not lead to particularly strong improvements, but we believe that our discussion in this submission serves relevant objectives of (a) kicking-off a discussion in the black-box label-shift literature, by pointing to the practical implications of some of the strong assumptions typically made in experimentation, and (b) providing practitioners with some encouragement about the fact that such methods can continue to be useful even when assumptions are violated (with simplifications when techniques don’t work out of the box), and potentially improved by taking advantage of extra degrees of freedom.
>
>  – To our knowledge, adding a tunable scalar and renormalizing is not common practice in the label-shift community when facing degenerate matrices. The widely-adopted practice so far in this literature has been to avoid such matrices altogether, mostly by making hold-out validation sets large enough, and working on well-balanced problems unlike the long-tailed dataset we use in iWildCam. The typical recommendation in the label-shift literature when encountering such confusion matrices has been to either use a pseudo-inverse or a soft-confusion matrix. We initially found these alternatives to be detrimental to downstream accuracies in initial experiments (we have added a set of results in Appendix E), which is why we opted for a “mixing” of the original confusion matrix with an identity matrix. However, we agree with reviewer feedback that this heuristic is somewhat ad-hoc, and has the limitation of adding an extra hyper-parameter. Furthermore, most often, we found the tuning process to pick the largest scalar, suggesting an asymptote towards an identity approximation. Therefore, we have revised our heuristic to simply use the identity, which we also believe makes for a smoother narrative, since our Bayesian view of FTH comes from this approximation.
>
>  – Our key intuition is that learning additional degrees of freedom for the “strength” of logit adjustment on an OOD validation set is more likely to reflect the level of correction that works best once a classifier is “out there”. When in-distribution, classifiers that rely on spurious correlations are high-performant, and learning scaling parameters on this set is more likely to generalize poorly in OOD settings. We have added results in Appendix D, Table 6, where we show how methods fare when hyper-parameters are selected using in-distribution/out-of-distribution/oracle validation. These numbers suggest that OOD validation is most often beneficial over in-distribution validation (and oracle validation of course works best).

---

### Official Review · Reviewer_9gjn · 2022-10-25

**Confidence:** 4
**Correctness:** 3
**Technical Novelty And Significance:** 2
**Empirical Novelty And Significance:** 2
**Recommendation:** 3

**Clarity, Quality, Novelty And Reproducibility:**

The paper, while being short and concise, is for the most part easily readable. Some sections that I found to be more difficult to understand are listed above.

Experiments are described clearly and seem reproducible. Some minor misunderstandings that I had regarding experimental protocols are listed above.

As far as I can tell, the paper's examination of online labels+conditional shift adaptation of neural networks is novel, as are the experiments performed here.

As detailed above, the quality of the paper can in my opinion be greatly improved if more context was provided about the distribution shifts considered, a more thorough empirical investigation was conducted, and the unclear/undefined terms and sections are clarified.

**Strength And Weaknesses:**

Major Strengths:

1. The problem investigated is well motivated. Distribution shifts are indeed a big and relevant problem when machine learning models are deployed in the real world. Much of the work to date has focused on idealized types of shifts, like label or covariate shift. It is natural to wonder how much methods developed for idealized shifts might be useful in more realistic settings. Furthermore, if label shift adaptation methods generalize to realistic shift scenarios, they are attractive from a practical standpoint, as they do not require labelled test data.
2. Empirical results are, for the most part (except for some comments below), clearly presented: I could understand what was done and believe I have enough information to attempt to reproduce the results.
3. The paper is quite honest about the inconclusive nature of much of the results, and does not try to oversell the proposed methods.

Major Weaknesses:

1. A systematic or principled approach to the types of distribution shifts considered is missing. Distributions can shift in many ways and for many reasons. Adding conditional shift to label shift is tantamount to considering general distribution shifts. Indeed, the paper considers two examples with no label shift (P(Y) is not changed in the synthetic MNIST and COCO-on-Places datasets), an example with covariate shift (Mixture of Gaussians), and two general distribution shifts (from the WILDS dataset). Framing the issue as “label shift in the presence of conditional shift” might give a wrong impression that the conditional shift is a perturbation of the label shift condition. I find it clearer to state that general distribution shifts are considered.
Little can be said about distribution shifts in general, without focusing on particular types or characteristics of the shifts, such as label/covariate shift, subpopulation shift [6], or shifts where the data generating process has a fixed known causal structure [3]-[5]. Since experiments in the paper do not belong to a particular type of shift, it is hard to compare results or to generalize from them to general shifts.
The lack of a systematic approach to general label shifts is reflected also in the absence of discussion of relevant work on this issue, including refs [1]—[6].

2. Given the vast scope of possible distribution shifts, with no systemic understanding of how they relate to or differ from label shift, and with heuristic methods lacking a theoretical foundation – given these, a major and comprehensive empirical study is necessary in order to ascertain the usefulness of the proposed methods. The paper offers modest experiments, in terms of types and strengths of shift, types of data, and alternative baselines/methods. This severely limits the usefulness of the results, as it is unclear when the suggested methods can be expected to improve upon baselines, and how good such improvement are compared to alternative methods. As it stands, few generalizable insights can be drawn from the empirical scope of the paper. The paper itself is honest about the modest and tentative nature of the findings, when it concludes that the experiments are “suggestive” that the proposed methods show “promising trends for the most part” in the limited scope in which they were tested.
Concretely, for the experiments performed, here are some suggestions of baselines/methods that might provide a wider context for obtained results:
  a. An optimal fixed classifier, as considered by Wu et al. (2021).
  b. Results obtained from offline domain adaptation methods (Garg et al., 2020).
  c. Results obtained from known domain generalization methods such as those mentioned in the related works section of the paper, or the ones surveyed by Gulrajani & Lopez-Paz (2020). In particular, if I understand correctly, CORAL was used for the two WILDS datasets considered in the paper, but not the others. It might be more informative to test all datasets with and without CORALS (and/or other domain adaptation methods).
  d. The paper emphasizes the importance of the use of an OOD validation set. It would thus be useful to test the effect of this OOD validation set on test performance by considering the effect of different validation sets, preferably with different characteristics. For example, for the synthetic colored MNIST dataset, one could use validation sets that are more or less correlated with the test sets.

3. Goals 2+3 above are not explored in detail in the paper. No references are given to prior work on regression label shift / domain adaptation (e.g., [7]-[8] below), nor to the discussion in Lipton et al. (2018, section 7) about remedies to non-invertible empirical confusion matrices. The corresponding experiments provide only an initial investigation into them. The paper provides some interesting but embryonic discussion/exploration of both. Their inclusion in the current form of the paper

4. Some key definitions and explanations are lacking in the paper, making it difficult to understand some sections of it.
  a. “Conditional shift” is not defined. While it is a term used in the literature and whose meaning might be intuitive, many other terms are used in the literature as well. To guarantee that there are no misunderstandings regarding this central concept, its definition should be provided.
  b. Method FTH-H-B and FTH-H-B (R) are never clearly defined (what is the “pseudo-count hyper-parameter” mentioned? I did not understand).
  c. In equation (3), the definition of the expected error rate \ell^{\test{new}} is only given in words, not in a formula.
  d. In section 4.1, what are a, b, kappa, and mu?
  e. In appendix A, none of the notation is defined, and in fact no information is given about the context and goal of the derivation there.

Further comments

1. Online vs offline methods. The scope of label shifts considered in this paper is more limited than those considered by Wu et al.: here only constant shifts are considered (test data is drawn from a fixed shifted distribution), whereas Wu et al considered distributions that keep changing throughout training. An important strength of online methods are their ability to deal with continual changes. Considering only constant changes reduces (but does not invalidate) the usefulness of online methods compared to offline ones. The decision to focus on online methods should be motivated in the paper.

2. OOD validation: the concept of OOD validation is introduced in Heuristic 1 without being properly defined/explained. As this is a central tenet in the proposed methods, the idea and procedure should have a clear and detailed explanation. Furthermore, in Heuristic 1 it is written that OOD validation is a standard practice of model selection, with a reference to Gularjani & Lopez-Paz (2020). As far as I can tell, this reference (which emphasizes the importance of validation set details in the context of domain generalization) does not advocate the use of validation on a separate OOD set. Rather, it attributes this method to Krueger et al. (2020), who indeed mention it in an appendix.
Regarding the method itself OOD validation itself: why should it work? I can understand that it might be useful when the shifts in the validation and test sets are somehow related (like the Skewed-MNIST example where test is a more severe shift of the same type as validation), but why would it help in examples like the mixture of Gaussians, or the WILDS datasets? Looking at the experiment results, it indeed seems to me that OOD validation is helpful only for the skewed-MNIST example. If my reading is correct, this should be stated clearly, and the appropriate qualifications should be made in the conclusions about the merits of OOD validation. Currently, section 5.3 states that “Using OOD validation sets … improves results on the whole” – but for S-COCO-on-Places and iWildCam (Avg) I do not see any improvement more significant than the noise level, and for iWildCam (F1) there is a small deterioration (which is also consistent with noise).
From a practical perspective, performing OOD validation is not always possible as it requires more labelled data – it would be useful to emphasize this fact. Technically – what are all the optimization steps performed on this validation set? I.e., which hyper-parameters are calculated on this validation set, other than the confusion matrix?

3. Non-invertible confusion matrices. The methods proposed in Heuristic 3 surely generate invertible matrices, but why would they be expected to work for label shift and general distribution shift adaptation? They seem to me ad-hoc and unmotivated. What would be their merit compared to using a pseudo inverse, or the soft probability matrix suggested by Lipton et al. (2018)?

4. Section 4: The role of this Bayesian discussion is not clear to me. What insights are gained from this Bayesian perspective? Are these insights relevant also to cases of pure-label shift, or only general distribution shifts? I found the discussion around equations (11)-(14) confusing on first reading. The notation in equations (11)-(12) is confusing, perhaps Y|\phi ~ Cat(\alpha) and \phi ~ Dir(\alpha). The notation in equation (13)-(14) – P_t(\phi), P^{new}_{t+1} is not defined anywhere.
I found the whole of section 4.1 confusing. How is the discussion related to label shifts in regression problems? What are the takeaways or results of this section? Are the results valid only for the Gaussian example with a conjugate prior, or more generally applicable? What kind of calibration is performed in this section, and why is it useful?

5. Experiment details. Right before section 5.1:
  a. It would be worthwhile to provide the details of “the surrogate loss implementation of Wu et al.”
  b. What are the details of the grid search used for the parameter of OGD? On which validation set is it taking place.
  c. Skewed-MNIST should reference the inspiration from color MNIST of Arjovsky et al. (2019). A table with the makeup (number of digits of each color) of each of the train/val/test datasets would be useful. It is stated that “Since the overall class frequencies are balanced … we drop the P(Y)”. Drop it from where (same comment for skewed COCO on Places)? Appendix C.1 describes how digits were split into two sets – was there a precise protocol for this? How is the “tend(ency) to be confused” measured context? What was the optimizer used for training  - SGD?
  d. WILDS-iWildCam: it is stated that “We use Heuristic 3 for evaluating methods on this dataset. Heuristic 3 mentions several approaches: adding a tunable scalar to the diagonal? Using the identity matrix? Using a “pseudo-count”?
  e. Table 2: How are the error estimates estimated relevant to all tables)? Why are the error estimates here +- 0? Are the quantities really measured to perfect accuracy?

Minor comments

1. Before equation (4): “where e is a one hot vector for the predicted category” – the description and notation there can be clarified: it was initially unclear to me which predicted category is referred to, and only after reading Wu et. al (2021) did I understand that these are calculated for each step I separately.

2. After equation (4), it is stated that calculating the gradients is tricky. Why is it so? For self-containedess, the statement should be explained. Similarly, before equation (7) it is stated that FTH is more efficient than OGD – efficient in which sense? Compute time? Memory? Data complexity?

3. Right before 5.1.3, it is mentioned that “test-sets are smaller”. Smaller than what?

4. Typos:
- Heuristic 1, line 2: shiftis -> shift is
- Two lines below equation (19): minimum -> minima
- Last line of page 6: there’s a superfluous ).
- 5.2.2, last line of first paragraph, should read “in neither training nor validation sets for OOD test.”
- The reference to Sun and Saenko (2016) is missing bibliographic info (journal name).
- Appendix A: equations (23) and (24) seem to be the same


References
[1] Storkey, When training and test sets are different, in:Quinonero Candela et al., Dataset Shift in Machine Learning, 2009
[2] Moreno-Torresa et al., A unifying view on dataset shift in classification  (2012)
[3] Schoelkopf et al., On Causal and Anticausal Learning (2012)
[4] Zhang et al., Domain adaptation under target and conditional shift (2012)
[5] Kull and Flach, Patterns of dataset shift (2014)
[6] Breeds: Benchmarks for subpopulation shift, Santurkar et al. (2020)
[7] Cortes and Mohri, Domain Adaptation in Regression (2011)
[8] Cortes and Mohri, Domain adaptation and sample bias correction theory and algorithm for regression (2014)


**Summary Of The Paper:**

It is well known that the performance of machine learning models is highly dependent on the distribution of the data on which it is evaluated: model performance deteriorates when tested on data generated from a distribution shifted with respect to the training data generating process. Identifying and mitigating the effects of distribution shifts is a major open challenge for machine learning practitioners, as distribution shifts are ubiquitous in an ever-changing world. In the supervised learning context, evaluating test performance and mitigating it usually require labelled testing data, which is often difficult or impossible to obtain.

Little can be done about arbitrary distribution shifts – generalization from training to test data is only possible if the shift leaves some structure in the data unchanged. Label shift is a basic example of such a distribution shift, where the conditional probability P(X|Y) remains fixed, and only P(Y) changes. Here X are the covariates, Y the label, and P(X,Y) = P(X|Y)P(Y) is their joint distribution. Recent years saw much progress with the analysis of label shift, and methods have been developed to mitigate its impact on black box models – with deep learning a primary application -- in both offline and online settings. Essentially, these methods rely on re-weighting model predictions using the distribution of predicted (pseudo-)labels, and thus do not require true labels for the test data.

The current paper follows three goals related to label shift adaptation:

1. The paper’s main effort focuses on examining how previously proposed label shift mitigation methods perform on shifted distributions do not satisfy the label shift condition – a scenario highly relevant to real-world applications, where often as pure label shifts are rare. In an online learning setting, albeit one in which the distribution does not shift continuously, the paper examines empirically how recently proposed algorithms for online adaptation to label shift perform on a few synthetic and realistic datasets that exemplify different kinds of “non-label” distribution shift. The empirical investigation also considers a couple of heuristically-motivated extensions to these algorithms, most notably performing model selection on an OOD validation set which is shifter with respect to both the training and test sets. The findings of these investigations are not clear cut, but suggest that in some cases, the proposed algorithms provide an improved adaptation to the distribution shift. The takeaway is that label shift adaptation methods (or some heuristic generalization thereof) might sometimes be useful to mitigate general distribution shifts, even if this practice has no known theoretical justification.

The paper considers two further issues related to label shift adaptation:

2. Past work on label shift adaptation has mostly focused on classification problems. The paper proposes an algorithm for label shift adaptation in regression settings, and studies it empirically.
3. Past algorithms for online label shift adaptation require the inversion of an empirically measured confusion matrix. The paper suggests a heuristic fix for the case when this matrix is non-invertible and studies it empirically.

The latter two issues are discussed briefly (compared to the main topic of the paper), and here too the investigations do not provide clear cut conclusions on the efficacy of the proposed methods, but in some cases these methods perform better than the baseline.


**Summary Of The Review:**

The work presented in this paper is novel, seems technically correct, and addresses a key problem to many real-world scenarios. I believe that the work in its current state with some corrections/improvements could and should merit publication in some venue. However, with the flaws described above, I do not believe this paper is ready for publication.

---

> ### Author Response · Authors · 2022-11-19
> **Response to review (major points)**
>
> Thanks for the review!
>
> > Major Weaknesses:
>
> > 1. Nature of distribution shifts; characterization as ‘general distribution shift’ more appropriate
>
> We agree that we are fundamentally dealing with general distribution shifts. In an earlier draft, we used the more common phrase in the domain adaptation literature, “generalized target shift” (GeTARS) from [4] (Zhang et al.) to refer to our problem setting. However, at the time, we received feedback that this term was confusing to readers, and that we ought to use a title and description that makes it clear that our adaptation is explicitly for the output-distribution-adjusting label-shift problem. Re. our examples with “no label shift”, we do perform adaptation separately in the test environments before aggregating accuracy, and in each of these environments, there is both label-shift and conditional shift (by design in the synthetic experiments, and as analyzed/suggested in Koh et al. for iWildCam and PovertyMap).
>
> > 2. More comprehensive experimentation would allow for clearer conclusions.
>
> We agree that conditional-shift can manifest in a variety of ways (correlation shifts, more-extreme covariate shifts such as domain shifts, subpopulation shifts), and we have not explored the full range. We also agree that for a really thorough empirical demonstration, we ought to explore the range of distribution-shift types at test-time as well as a range of base models likely to be encountered in reality, and also ranges/types of distributional-shift in validation sets. We do believe that our discussion in this submission serves relevant objectives of (a) kicking-off a discussion in the black-box label-shift literature, by pointing to the practical implications of some of the strong assumptions typically made in experimentation, and (b) providing practitioners with some encouragement about the fact that such methods can continue to be useful even when assumptions are violated (with simplifications when techniques don’t work out of the box), and potentially improved by taking advantage of extra degrees of freedom.
>
> We have added results in Appendix D, Table 6, where we compare hyper-parameters selection with IID/OOD/oracle validation. These numbers suggest that OOD validation is most often beneficial over IID validation (and oracle validation of course works best). As suggested, we have added the Optimal Fixed Classifier (OFC) in Table 1, and a variant OFC-H, where we learn scaling hyper-parameters on top of the fixed p.
>
> > 3. Discussion about degenerate confusion matrices embryonic; advantages unclear over existing alternatives.
>
> We agree our discussion of confusion matrices was not very detailed. As further background, we developed Heuristic 3 when we found use of both the pseudo-inverse as well as the soft-confusion matrix to result in poor performance on the iWildCam dataset. We found that as we added larger values to the diagonal of the confusion matrix and renormalized (i.e. converged to the identity matrix), downstream validation performance recovered. We tuned the additive scalar on an average of downstream OOD validation accuracy for FTH and OGD. However, we note that most often, the largest value was picked through this scheme, suggesting that the identity-approximation is usually better. Given overall reviewer sentiment about the ad-hoc nature, and the limitations induced by adding yet another hyper-parameter to the system, we are rolling this back to simply using the identity matrix itself as an approximation. This also makes for a smoother narrative, since our Bayesian re-interpretation for FTH is derived from the identity approximation. We have added a discussion in Appendix E (Tables 7 and 8).
>
> Regression problems have indeed been considered generally in the literature on adaptation, and we are adding in the suggested citations. To our knowledge, a specific discussion for online black-box label-shift has not been discussed in the context of regression, analogous to the discussion for classification problems in Wu et al. (2021). We have clarified this in the draft.
>
> > 4. Lack of clarity in discussion about conditional shift, notation, and other points.
>
> In this submission, we generally consider the “conditional shift” that occurs due to changes in location; the implication is that the semantic features conditioned on labels do not change significantly, but that background features can, due to the environmental shift. Our notations and discussion in Section 4 are heavily based on the literature, in particular that in our citations of DeGroot (2004) and Murphy (2007). We apologize for not adding descriptions around the derivation in Appendix A; the goal is to show why (and under what conditions) the Bayesian posterior-update equation is valid (it is valid when the samples in a new deployment location are drawn independently in that location). We have edited our draft to reflect these clarifications.

---

> > ### Author Response · Authors · 2022-11-19
> > **Response cont'd (Other points)**
> >
> > > Further comments
> >
> > > 1. Motivate online vs. offline methods. Discussion restricted to constant changes.
> >
> > We agree with these points; our motivations for this work were primarily based on the use of cloud based APIs in real-time deployment in several locations, necessitating the black-box, post-hoc, online setup. Such deployments are often across several clients in diverse locations with unique label-distributions, as captured in our choice of WILDS datasets. Such deployments often come with additional resource-constraints, since ideally one would perform client-based adaptation on-device, which is a good use-case for cheap approaches based on output-distribution-adjustment. While temporal drift would indeed be a natural extension for such online applications, in this submission we restrict our scope to exploring constant-shift.
> >
> > > 2. Further discussion of intuitions behind heuristics required. What hyper-parameters are picked on the validation set?
> >
> > We agree that OOD validation should not be expected to help when the nature of the distributional shift is significantly different between validation and test-time settings. Model-selection is quite a fundamental problem in OOD settings with no obvious solution. Regarding the specific methods we consider, our intuition is that learning additional degrees of freedom for the “strength” of logit adjustment on an OOD validation set is more likely to reflect the level of correction that works best once a classifier is “out there”. When in-distribution, classifiers that rely on spurious correlations are high-performant, and learning scaling parameters on this set is less likely to generalize OOD. To illustrate this, we have added results showing the performances when performing validation using IID/OOD/Oracle sets in Table 6 in the Appendix.
> >
> > The learning rate for OGD-based methods, and the scaling hyper-parameters for FTH-H and OGD-H (and also the pseudo-count for FTH-H-B) are picked using the validation sets.
> >
> > > 3. Ad-hoc heuristic for generating invertible confusion matrix.
> >
> > Please see above, response to Major Weaknesses, point 3.
> >
> > > 4. Role of the Bayesian discussion, notation, discussion on regression unclear.
> >
> > The Bayesian interpretation of FTH leads us to the FTH-H-B method for classification problems, and the generality of the update equations helps us easily derive analogous updates for regression. Such insights ought to be equally applicable to pure-label shift. We agree our notations were awkward in (11-12), we’ve made changes. Section 4.1 tells us how the output distribution in a regression problem might be similarly “reweighted” to account for label-shift in a new location. The equations are generally applicable, however, one ought to use conjugate priors for whatever likelihood model one picks, otherwise the integral in Eq. 14 becomes intractable (although one might also attempt approximations here). By “calibration” we simply meant the scaling hyper-parameters, playing a role equivalent to those in the classification problems, and they are useful similarly.
> >
> > > 5. Miscellaneous comments and clarifying questions (a)-(e).
> >
> > We apologize for not being clearer. (a) We have added a brief description for the surrogate loss implementation; (b) The grid search was conducted for the learning rate in OGD from 1e-8 to 10.0 in steps of x10; (c) We’ve added these details. Re. P(Y) being dropped, we drop it from the decision rule, since weighting by a uniform distribution does not change the rule. Re. the protocol for the S-MNIST dataset, we trained the same base network the same way as in the experiments (SGD with weight decay) for 200 iterations, and then looked at the confusion matrix on the test set. This tells us which digits get confused the most, and we used this partition the digits such that each split contains 5 digits; (d) Heuristic 3 does not involve a pseudo-count, it only deals with the approximation for the confusion matrix. We used the tunable-scalar variant in our submission, but we have changed this to simply be the identity matrix, as described above; (e) In Table 2 for PovertyMap, the standard deviations are over multiple re-orderings of the test-sets in each location (since we are evaluating an online method). The error estimates are very low because the methods we evaluate are generally quite robust to random re-orderings. In other cases, when we do not split up results by folds or over separate trainings of the base network, the variation comes to a large extent from the variation in the training of the base networks (as can be partially inferred from the standard deviations for the Base results). We observed all methods to be very robust to random re-orderings of the test sets, except for S-COCO-on-Places, which is why we aggregated results over 20 trials for each seed.
> >
> > > Minor suggestions for writing/typos.
> >
> > We have updated the draft fixing these.
> >
> > Once again, thank you for the very thoughtful, well-written, and constructive review!

---

### Decision · Program_Chairs · 2023-01-20

**Decision:**

Reject

**Justification For Why Not Higher Score:**

I would like to thank the authors for their replies to the questions raised by the reviewers and for updating the manuscript, which improved the understanding of the authors' work to some extent. However, the overall contribution of this work is limited, in particular, lack of theoretical justification is critical. Therefore, I cannot recommend the acceptance of this paper.

**Justification For Why Not Lower Score:**

N/A

**Metareview: Summary, Strengths And Weaknesses:**

Summary:
This paper discusses the problem of online learning with label shift in the presence of conditional shift.

Strength:
The paper is well written and the investigated problem is general and well-motivated.

Weakness:
The authors' proposal is heuristic and there is no theoretical justification.